# Proteomic and Metabolomic Changes in Psoriasis Preclinical and Clinical Aspects

**DOI:** 10.3390/ijms24119507

**Published:** 2023-05-30

**Authors:** Adrianna Radulska, Iwona Pelikant-Małecka, Kamila Jendernalik, Iwona T. Dobrucki, Leszek Kalinowski

**Affiliations:** 1Department of Medical Laboratory Diagnostics—Fahrenheit Biobank BBMRI.pl, Medical University of Gdansk, 7 Debinki Street, 80-211 Gdansk, Poland; adrianna.radulska@gumed.edu.pl (A.R.); iwona.pelikant-malecka@gumed.edu.pl (I.P.-M.); kamila.jendernalik@gumed.edu.pl (K.J.); 2Beckman Institute for Advanced Science and Technology, University of Illinois at Urbana-Champaign, 405N Mathews Ave., MC-251, Urbana, IL 61801, USA; 3Department of Bioengineering, University of Illinois at Urbana-Champaign, Urbana, IL 61801, USA; 4Carle-Illinois College of Medicine, University of Illinois at Urbana-Champaign, Urbana, IL 61801, USA; 5BioTechMed Centre/Department of Mechanics of Materials and Structures, Gdansk University of Technology, 11/12 Narutowicza Street, 80-233 Gdansk, Poland

**Keywords:** in vivo mouse models, psoriasis, proteomics, metabolomics

## Abstract

Skin diseases such as psoriasis (Ps) and psoriatic arthritis (PsA) are immune-mediated inflammatory diseases. Overlap of autoinflammatory and autoimmune conditions hinders diagnoses and identifying personalized patient treatments due to different psoriasis subtypes and the lack of verified biomarkers. Recently, proteomics and metabolomics have been intensively investigated in a broad range of skin diseases with the main purpose of identifying proteins and small molecules involved in the pathogenesis and development of the disease. This review discusses proteomics and metabolomics strategies and their utility in research and clinical practice in psoriasis and psoriasis arthritis. We summarize the studies, from in vivo models conducted on animals through academic research to clinical trials, and highlight their contribution to the discovery of biomarkers and targets for biological drugs.

## 1. Introduction

Psoriasis is a common and recurrent immune-mediated disease, mainly manifested by skin lesions of well-demarcated and erythematous plaques usually covered with silver scale. The presence of this pathological feature results from epidermal hyperproliferation, abnormal keratinocyte differentiation, neovascularization, and extensive inflammatory infiltration. The etiology of psoriasis is multifactorial and still insufficiently known, but the dominant role is indicated by the dysregulation of the immune system with genetic susceptibilities. Furthermore, a variety of environmental factors, such as stress, infections, smoking cigarettes, and alcohol consumption, are also widely related to the development of psoriasis [1]. The current worldwide prevalence of psoriasis among adults ranges between 0.11% to 6.10%, with geographical and populational discrepancies [2]. Higher incidence rates have been reported in Western and Central Europe and North America, whereas the lowest rates are in Africa and Asia. The occurrence of psoriasis among the Polish adult population is estimated at about 2.99% and was more prevalent in the north than in the south of the country [3,4]. Epidemiological data from many countries are very limited or unavailable, therefore, these values may be underestimated [5]. Recent studies have suggested a constant or insignificantly decreasing tendency in psoriasis incidence but an increase in their prevalence [2,6]. Generally, psoriasis has no gender predilection, but some studies have shown that women are more likely to develop the disease than men. First symptoms may occur at any age, however, some researchers suggest two peaks of onset—in early adulthood and in the second part of life. Additionally, factors such as age, race, and gender contribute to the diversity in the prevalence of psoriasis [2].

Psoriasis is a heterogeneous disease with several different clinical types depending on the morphology and anatomical location of lesions. The most widespread and well-recognized type of psoriasis with its skin manifestation is vulgaris (PsV) or plaque psoriasis (PsP). Several other rare clinical psoriasis subtypes have been described, such as guttate, erythrodermic, or pustular psoriasis, but their frequency of occurrence is less than 10% of cases [7]. The extracutaneous manifestation of psoriasis is psoriatic arthritis (PsA), which leads to chronic, systematic inflammation and debilitation of joints, ligaments, and tendons. Psoriatic arthritis affects 30–35% of patients with dermal subtypes of psoriasis as a result of an exacerbation of the disease [8]. Psoriasis is strongly associated with the development of comorbidities, such as cardiovascular, metabolic, and mental health diseases, often referred to as systemic diseases [9]. The variety of clinical symptoms and the severity of the disease requires complex diagnostics and treatment regimens, which is a great challenge for clinicians. Standard clinical procedure is based on the physical examination of the skin lesions. The hallmark of psoriasis is skin plaques that are usually located on the scalp, trunk, and limbs, especially in the flexural area, but in some cases, the nails are also involved. The hallmark of plaque psoriasis is red, demarcated plaques that are usually covered by silvery scales. The detachment of the adherent scales can lead to the appearance of punctate bleeding spots, known as the Auspritz’s sign. Plaques are usually located symmetrically on the scalp, trunk, and limbs, especially on the elbows and knees. Severe forms of psoriasis can also lead to nail involvement. The characteristic feature of patients with psoriasis is the presence of the Koebner’s phenomenon, which is the appearance of new lesions on previously unchanged skin areas as a result of trauma. Even minor exposure to triggers, such as skin injuries, infections, or tattoos can trigger the development of Koebner’s phenomenon [10,11]. In doubtful and atypical cases, a skin biopsy can be performed, but it is not a routine procedure. Recommended laboratory tests, such as C-reactive protein or red blood cell sedimentation rate (ESR), are mainly used to monitor inflammation and indirectly predict the effectiveness of treatment. Nonetheless, available diagnostic tests are insufficient; therefore, the search for specific biomarkers at the early stage of the disease development is needed. For several decades, scientists have been trying to find clinically relevant biomarkers that will differentiate psoriasis subtypes to aid in early diagnosis and monitor the effectiveness of treatment. A precise understanding of the roles of pathogenic inflammatory and molecular pathways in psoriasis has enabled the use of highly effective, targeted biological therapies for treatment [12]. Nowadays, one of the most pivotal directions of research is metabolomics and proteomics. This review discusses proteomics and metabolomics strategies and their utility in research and clinical practice in psoriasis and psoriasis arthritis. We summarize the studies, from in vivo models on animals through academic research to clinical trials, and highlight their contribution to the discovery of biomarkers and targets for biological drugs.

## 2. Rodents as a Model for Psoriasis

The currently available animal research models for psoriasis do not fully reflect human pathophysiology; However, they can be a powerful and potential tool for preclinical application and biomarker research. Based on mice, a common rodent used in research, researchers have both identified psoriasis spontaneous models and created models using advanced techniques, such as xenografts or genetic engineering.

### 2.1. Spontaneous Mouse Models

The first psoriasis-like dermatitis model has been identified as spontaneous mouse mutation in the BALB/c strain, characterized by, e.g., hyperkeratosis, alopecia, and single hair-follicles [13]. Mice with spontaneous genetic mutations, such as Asebia (AB), flaky tail, or flaky skin (Fsn), possess psoriasis-like symptoms that are not fully comparable to humans, including parakeratosis, acanthosis, changes in vascularity, and infiltration of mast cells. However, lack of T-cells has been observed in infiltrates in psoriasis-like lesion spontaneous mouse models [14,15,16,17,18,19]. The interleukin 17 family plays a crucial role in the pathogenesis of human psoriasis. Neutralization of one of the members—IL-17C—has the effect of reducing skin inflammation in the flaky tail mice model [20]. Moreover, studies highlight a potential role of IL-17C in psoriasis inflammation as a mediator, where it stimulates Th17 cells through signaling using heterodimeric receptor IL-17RC, which possesses an IL-17RA and IL-17RE unit [20,21,22]. Data suggest that IL-17RA could be a potential biomarker of treatment response in psoriasis (in reference to an interactive map of biomarkers in psoriasis: (https://imi-biomap.elixir-luxembourg.org/minerva (accessed on 31 January 2023)). Considering the above spontaneous mouse models shows them to be a potent research tool, not only in basic aspects of psoriasis but also in searching for potential biomarkers.

### 2.2. Induced Modulation of the Skin Environment

Changes in the skin environment can be reached by repeated topical application of imiquimod (IQ) or dermal injection of IL-23. The first model is the most common and convenient to use in the psoriasis-like inflammation mice model. IQ is a commercially available Toll-like receptor (TLR) 7/8 agonist which can be used for testing the therapeutic potential of a new drug or searching for psoriasis disease mechanisms. Pathogenic psoriasis disease underlines the significant role of IL-23 in the development and progression of this disease [23]. The in vivo model of acute inflammation in IQ mice manifested elevated levels of IL-23 and showed mediation throw axis IL-23/IL-17, which is similar to human psoriasis [24,25,26,27,28]. A study with the IQ mice model confirms the anti-inflammation effect of a competitive angiotensin II receptor antagonist (azilsartan, a commercially available drug) with a simultaneous decrease of IL-23 serum level [26]. At the same time, another protein—keratinocyte derived 2-5-oligoadenylate synthase 2—has been selected as a potentially sensitive biomarker for severity prediction of psoriasis, which has been reflected in the IQ mice model [29]. The summarized IQ mice model, which reflects changes in biomarkers such as IL-23 during psoriasis development, has the potential for research application [30]. The mice model of dermal injection of IL-23 allows researchers to study pathways in the development of psoriasis [31]. Dermal injection of psoriasis biomarker IL-23 directly into mice results in the development of skin inflammation driven by Th17 cells (the same as in human psoriasis) and upregulation of pro-inflammation cytokines (IL-22, IL-17A, INF-γ) [32,33,34]. Simultaneous dermal injection of IL-23 with a combination of, e.g., in mice IL-6^−/−^ shows that the development of psoriasis driven by IL-23 is mediated by other essential interleukins, such as IL-6 [32].

### 2.3. Psoriatic Human Skin Xenograft Model

Xenotransplantation of a human psoriatic skin mouse model with immunodeficiency has been widely used for new anti-psoriasis drug research [35,36]. This mouse model possesses preserved immunological and phenotypic properties of human psoriasis [37]. Active T-cells in donor grafts allow for the analyzed activity of candidates for new anti-psoriasis drugs, which target T-cell pathways [37,38]. It should be mentioned that, in this model, T-cells activity in donor grafts decreases over time [39,40]. Moreover, data show that active NK cells, which have normally been active in scid/scid mice, have been responsible for graft rejection [41,42]. Another mice model, AGR 129, has a great advantage over scid/scid mice. According to data, human xenografts transplanted into AGR 129 mice recipients have not been rejected because of immature NK cells observed in this model [43]. During the development of psoriasis phenotype in AGR129 mice after xenotransplantation, upregulation of markers such as intercellular adhesion molecule 1 (ICAM-1), MHC class II, endothelial cell adhesion molecule-1 (PEACAM-1) and pro-inflammation cytokines (TNF-α, Il-12 and IFN-γ) [43] has been observed. It should be mentioned that ICAM-1 has been considered a biomarker of psoriasis, which is downregulated in patients treated with golimumab [44]. To summarize, xenograft models have the potential not only to test new drugs and observe visible changes in skin condition but also to analyze changes after drug treatment at the molecular level using psoriasis biomarkers.

### 2.4. Transgenic Models

Transgenic mouse models (TM) are typically characterized by specific genetic changes that result in overexpression or knockout (KO) of a defined protein. In standard TM models, these changes can be observed in all cell types throughout the mouse’s body. However, with advancements in engineering techniques, it is now possible to restrict these genetic modifications to specific tissues in the rodent’s body. Furthermore, these genetic modifications can be controlled by specific gene promoters or gene expression modulators, such as tamoxifen or tetracycline (doxycycline) [45,46,47,48,49]. Those chronic inflammation models with psoriasis-like changes in mice skin condition can include acanthosis, hyperparakeratosis, altered keratinocyte differentiation, epidermal hyperplasia, hypervascularity, and mixed inflammatory infiltrate. Despite the fact that TM does not fully correspond to human psoriasis on histological and immunological levels, during past years, genetic mouse models have been used to understand the role of cytokines, factors, and inflammatory mediators in the development of psoriasis, which can be potential biomarkers of psoriasis [49]. Cytokines such as IL-17 and IL-23, which play a crucial role in the pathogenesis of psoriasis, have at least a few corresponding transgenic KO mouse models [45,50,51]. Moreover, mice models with molecules overexpression, such as vascular endothelial growth factor (VEGF), transforming growth factor α (TGF-α), IL-6, IL-1α, interferon gamma (INF-γ), bone morphogenic protein (BMP)-6, tyrosine kinase, kallikrein-related peptidase 6, and many others, have been used not only to clarify the role of a biomarker in psoriasis but also to analyze effects of new drug treatments or differences in disease progression [31,45,46,47,49,50,51,52,53,54,55].

### 2.5. T-Cell Transfer Rodents Model

Shifting immunological balance can result in the development of psoriasis-like symptoms in mice models. Models based on T-cell transfer are more difficult to use and are affected by the genetic properties of the recipient. Typically, T-cells from HLA-B27 transgenic rats [56], CD+/CD45RBhi T-cells (naive T cells, e.g., from B10.D2 mice) [57], or Th17 cells from desmoglein 3-specific–tg mice (Dsg3H1-Th17) can be used [58]. As recipients, immunocompromised nontransgenic rats [59], scid/scid mice [56,60], or mice that produce no mature T cells or B cells (recombination activating gene 2, Rag2^−/−^ KO mice) can be used [61]. Shifting the immunological balance using T-cells gives the opportunity to analyze changes in immune cell response and pathways involved in these processes. Moreover, this transfer model gives the opportunity to confirm that both Th-17 cells and related cytokines, such as IL-17 and IL-23 (biomarkers of psoriasis), are involved in the pathogenesis of psoriasis [61]. In summary, these types of models allow confirmation that IL-17 and IL-23 could be potential biomarkers of psoriasis development.

### 2.6. Genome Editing-Models Based on CRISPR/Cas9 Technology

A new biotechnological tool, the Clustered Regularly Interspaced Short Palindromic Repeats associated protein 9 (CRISPR/Cas9), changed the perception of genetic engineering of psoriasis in mice models. This Nobel prize winning technology allows gene editing in eukaryotic cells [62,63]. Transgenic mice with desmogelin 1 knockout exhibited peeling and denuded skin, typical for skin inflammation in the psoriasis-like type [64,65]. Another set of experiments with Tip1flox/flox mice showed that Tnip1 gene, which codes TNFAIP3-interacting protein 1, has an impact on the regulation of IL-17 [66]. Mice models with IL23A gene deletion restricted to keratinocyte confirm the signification of keratinocyte-derived IL-23A in psoriasis development and show the importance of IL-23 as a biomarker of disease [67]. In summary, models created with CRISPR/Cas9 technology are a potential tool to discover aspects of psoriasis pathogenesis.

Despite the wide selection of mouse models used in psoriasis research, none of them fully correspond to the natural pathophysiology of human psoriasis. Therefore, choosing a mouse research model still requires careful consideration of the benefits and drawbacks of the selected model (as summarized in Table 1). However, despite the limitations of these mouse models (including differences in skin structure, lifespan, metabolism activity and immune system response) [68,69] all of them can still be used to identify psoriasis biomarkers for future research in human psoriasis.

## 3. Proteomic and Metabolomic in Biomarker Discovery 

Recently, proteomics and metabolomics have been intensively investigated in a broad range of skin diseases. The main purpose is the identification of proteins and small molecules involved in the pathogenesis and development of disease. Here, we present proteomics and metabolomics strategies and their utility in research and clinical practice in psoriasis and psoriasis arthritis (Figure 1A). We distinguish two main strategies: non-targeted and targeted analysis. Strategy selection depends on several factors, such as instrumentation, data processing software, type of research, sample type availability, laboratory capabilities, and specialized staff. The limiting factor in the non-targeted proteomic and metabolic analysis is the availability of high-resolution mass spectrometers based on Time-Of-Flight (TOF) with electrospray ionization (ESI) or matrix-assisted laser desorption ionization (MALDI), or as hybrid instruments coupled with quadrupole (qQTOF) or ion traps (Orbitrap) [73,74]. Manufacturers are competing to provide more sensitive and efficient devices. High sensitivity and resolution mass spectrometers give the power for non-targeted ‘shotgun’ analysis used mainly by researchers. The proteomic screening in psoriasis was performed using iTRAQ [29,75,76] or TMT labeling [77] and non-labeled strategies [78,79,80,81], obtaining relative quantitation results of differentially expressed proteins (DEPs) as the difference between the control and psoriasis group. Identified biomarkers in psoriasis and psoriasis arthritis mainly belong to a few function categories. Most explored biomarkers are concerned with systemic inflammation, acute immune response (e.g., IL-6, IL-23, anti-factor VIII, and immunoglobulin GCT-A3) [78], cytoskeletal (profilin-1, kallikrein-8, component C3) [82], and Ca^2+^-binding proteins (S100A7, S100A8, S100A9) [80,82,83] in plasma. Palvina et al. applied the intact low molecular weight peptidome in psoriasis plasma to evaluate the concentrations of endogenous peptides, which allowed them to evaluate proteolytic activity. The results of peptidome analysis agreed with the protein identified in proteomic analysis. Increased concentrations of cytoskeletal proteins and their peptides in psoriatic plasma emphasize the combability of these two strategies in biomarker discovery [83]. It is worth pointing out that sample preparation in peptidomic methods is less challenging, and the method is easier to convert for quantitative analysis. However, the method’s equivalence still needs to be validated, and the clinical value of the identified proteins and peptides must be determined in longitudinal studies of psoriasis. Peptidomic analysis was also applied for the identification of disease-related peptides. Comparison of Ps, PsA, and control subjects (HC) showed significant differences in some peptides between Ps + PsA and HC groups. They identified the difference in the number and intensity of peptides between Ps and PsA, which could be used for the diagnosis of disease progression and differentiation [84]. Several reports focused on proteins involved in lipid metabolism and in the management of vitamin D levels in psoriatic and psoriasis arthritis patients [78,85]. Gegotek et al. investigated the changes in keratinocytes and lymphocytes in the blood and skin of psoriasis patients. In keratinocytes, they observed changes that occurred in proteins involved in calcium-binding, nucleic acid binding, and proteins with hydrolase activity. In lymphocytes, proteins responsible for nucleic acid binding and hydrolysis and proteins with oxidoreductase and enzyme modulator activity were observed. They also investigated the difference in the level of 4-HNE-protein adducts, where the shift from mainly 4-HNE-Lysine in control subjects to 4-HNE-Histidine, 4-HNE-Cysteine, and 4-HNE-Lysine at balance were observed in skin keratinocytes. A twofold increase in 4-HNE-protein adducts was also visible in lymphocytes, which slightly shifted from 4-HNE-Cysteine into 4-HNE-Histydyne adducts formation. 4-HNE is a biomarker of lipid peroxidation and lipid mediator that has an impact on inflammation and immunological responses. [79]. Most explored biomarkers of skin tissue belong to the skin barrier function and immune response, antigen processing, and IL-17 signaling pathways (e.g., kallikreins-8, S100A8, S100A9, component C3, SART 1, and GLTP) [75,80,86,87]. Some proteomic studies underly the potential of markers of connective tissue remodeling (e.g., MMP3 and M-CSF), implying these are specific to PsA [87]. In some reports, apart from untargeted studies, authors validate selected proteins by parallel reaction monitoring (PRM). Yu et al. identified thirteen proteins involved in two drug metabolism pathways (by cytochrome P450 pathway or other enzymes) in psoriasis vulgaris (PsV) subjects. Nine proteins (MPO, TYMP, IMPDH2, GSTM4, ALDH3A1, CES1, MAOB, MGST1, and GSTT1) were significant and were proposed for future evaluation as potential biomarkers of PsV pathogenesis and investigated by immunotherapy techniques [77].

Most metabolomic studies have been performed as untargeted discovery screenings to identify small molecules involved in psoriasis pathologies [88,89,90,91,92,93]. Obtained results underline the importance of polyunsaturated fatty acids and amino acid levels, which were involved in several metabolic pathways, such as immune activity, urea cycle, cell turnover, cardiovascular disease, nitric oxide synthase, collagen synthesis, or protein synthesis. Targeted metabolomics based on specific metabolite analysis is popular in some classes of metabolites, such as lipidomic [94,95,96] or amino acids [97,98,99]. Only a few reports present comprehensive data from untargeted discovery studies with results verification in targeted analysis. Non-targeted metabolomics was performed to identify significantly altered amino acids and carnitines in psoriasis patients. Based on that, carnitine and amino acid-targeted metabolomic profiling were investigated in plasma samples of mice induced by imiquimod (IMQ) to explore the role of metabolism in psoriasis. Studies identified 23 upregulated amino acids, including essential amino acids (EAAs) and branched-chain amino acids (BCAAs), whereas glutamine, cysteine, and asparagine were significantly downregulated. In the carnitine-targeted metabolomic analysis, 40 significantly altered carnitines were identified. Hexanoylcarnitine (C6) and 3-OH-octadecenoylcarnitine (C18:1-OH) were significantly upregulated, and 14 carnitines, included palmitoylcarnitine (C16), were downregulated in psoriasis [99]. The lipidomic analysis demonstrated changes in phospholipid profiles associated with changes in fatty acids composition. Ambrożewicz et al. showed the decreased concentration of phospholipid LA (18:2, 18:3), free AA (20:4) and DHA (22:6) in Ps and PsA patients compared with healthy subjects. In PsA, fatty acid levels were significantly reduced compared to Ps [95]. Another study used metabolomic fingerprinting to identify psoriasis conversion to PsA and disease activity biomarkers. Similar metabolomic profiles of psoriasis patients who developed PsA and mild PsA were observed. Some eicosanoids were detected only in samples of patients with moderate and severe PsA, and these eicosanoids are known to have pro- or anti-inflammatory properties. Patients with severe PsA had elevated levels of some long-chain fatty acids, 3-hydroxydodecanedioic acid, and 3-hydroxytetradecanedioic acid, which suggests dysregulation of fatty acid metabolism. Furthermore, 1,11-undecanedicarboxylic acid was identified as a classifier in PsA patients vs. healthy individuals [93]. Intensive metabolome research was also performed by Kishikawa et al. to identify metabolite biomarkers of psoriasis and its subtypes. They showed an increased level of ethanolamine phosphate and decreased concentration of XA0019, nicotinic acid, and 20α-hydroxyprogesterone in psoriasis samples than in the controls. In psoriasis subtypes (PsA vs. cutaneous psoriasis PsC), tyramine level was decreased, whereas mucic acid increased in PsA [89]. Targeted metabolomic analysis was performed for TMAO, arginine and their dimethyl analogs (ADMA), and homocysteine, which are cardiovascular biomarkers, to investigate comorbidities in psoriasis [90,97,98]. They found an increased concentration of TMAO and betaine, which correlate positively with PASI. Significantly higher levels of ADMA and homocysteine with lower citrulline and L-arginine/ADMA values in psoriatic patients was observed. They also presented a strong relationship between ADMA levels and disease severity in psoriasis patients.

Proteomic/metabolomic studies are very expensive because of the biological material collection (transport, storage), and procedure for sample preparation and analysis. Each step needs special consumable material, reagents, and analytical devices. Complex data analysis using special proteomics software and libraries programs needs time and specialists to reject the false positive results. The biggest advantage of non-targeted analysis is the verification of possible diagnostic markers and therapeutic targets, as well as molecular mechanisms and signaling pathways in skin disease development. From a thousand proteins/ molecules identified as potential biomarkers, only a few of them have the chance to be approved in clinical practice. Most publications present proteomic/metabolomic data using relative quantitation methods to determine protein/molecule expression as up or downregulated without the level of physiological norm limits. This could be crucial for final biomarker clinical validation.

**Figure 1 ijms-24-09507-f001:**
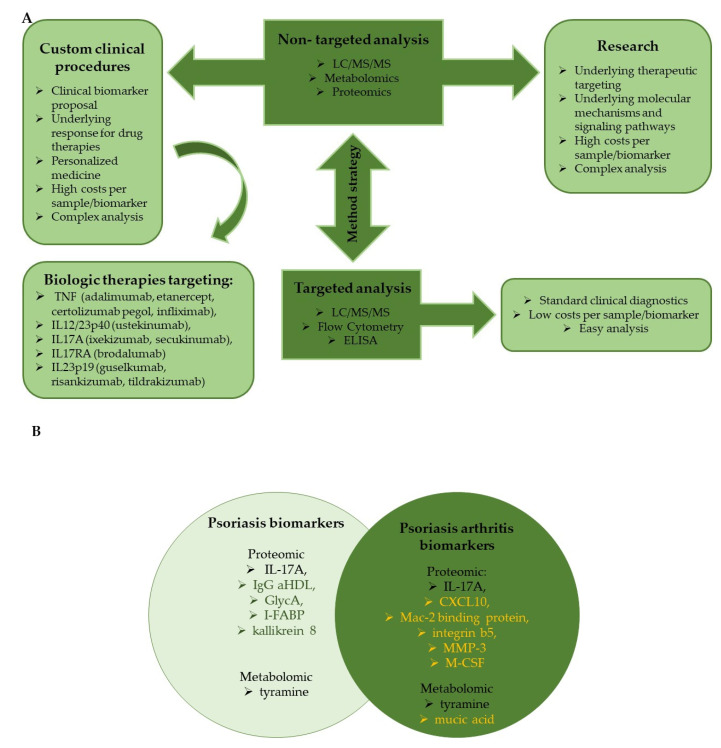
(**A**) Basic techniques used in proteomic/metabolomic analysis and their utility in psoriasis. (**B**) Catalogue of future candidate proteomic and metabolomic biomarkers differentiating psoriasis and psoriatic arthritis [100]. Black font represents biomarkers identified in both diseases and green/yellow fonts correspond, respectively, to Ps and PsA.

The international experts from the Biomarkers in Atopic Dermatitis (AD) and Psoriasis (Ps) BIOMAP consortium proceed with a cross-sectional two-round survey to predict the core elements of high-quality AD and Ps biomarkers to prepare medical recommendations [101]. The study underlines the importance of three statements: performance, purpose, and obstacle. The authors emphasize the importance that biomarkers of skin disease should characterize high-test reliability and high clinical performance. The therapeutic response statement was considered to be the highest purpose. The biggest challenges and priorities in biomarkers development were stated for more validation studies, harmonization, and creation of data sources [101].

Data comparisons between different study groups must be completed very carefully. Different instruments, study groups, biological material, sample preparation, analytical and acquisition methodology or data processing, programs, and statistical analysis are usually used. This all has an impact on the final results of non-targeted proteomic/metabolomic analysis, and it is challenging to harmonize workflow to obtain reproducible results. The next step should be method modification for targeted analysis, where the use of internal standards will allow for quantitative analysis. In this case, multi-reaction monitoring (MRM) workflow, internal standard correction, and calibration curves enable method standardization and validation for standard clinical analysis. The targeted analysis is much less costly and time-consuming, providing fast results for patients. The targeted metabolomics approach is much easier to implement because of the availability of metabolite standards and internal standards. This approach is often used in psoriasis and psoriasis arthritis analysis of lipids [94,95,96] and amino acids [97,98,99]. The targeted proteomics approach is more complicated to implement because of the low availability of internal standards and their high costs. Only a few reports are available that used qualitative analysis in psoriasis or psoriasis arthritis [29,76,77].

The latest publication summarizing current proteomics and metabolomics knowledge of biomarkers with future potential utility for predicting psoriasis severity and psoriasis arthritis was prepared by the international experts BIOMAP consortium. Presented results were based on 181 reports including only studies with more than 50 participants. They analyzed studies published mostly in the last decade, in which almost 49% were dominated by studies of proteomic biomarkers. In summary, of the studies of biomarkers in psoriasis and psoriasis arthritis, around 60% of them concentrated on the immune system, especially in cytokines and chemokines and acute case response, and less in immune cells and their signaling, antigen presentation, and innate immune response. Similar interest was taken (around 15–20%) in metabolism (mainly fat and iron metabolism), tissue homeostasis (angiogenesis, tissue remodeling, and skin barrier function), and intracellular signaling (hormonal signaling). A comparison of available data from 181 studies allowed authors to identify of six proteomic and two metabolomic biomarkers in psoriasis development. Interestingly, only one shared proteomic (IL-17A) and metabolomic biomarker (tyramine) was identified in both disease types. Biomarkers that could differentiate Ps vs. PsA were: IgG aHDL, GlycA, I-FABP and kallikrein 8, CXCL10, Mac-2 binding protein, integrin b5, matrix metalloproteinase-3 and macrophage-colony stimulating factor (proteomic), and mucic acid (metabolomic), which are presented in Figure 1B [100]. Half of the biomarkers presented above were discovered during the proteomic/metabolomic studies investigated by mass spectrometry, and half by standard analytical methods such as ELISA, western blot, flow cytometry, and NMR, which were often used for biomarker verification and validation in mass spectrometry reports. One of the important roles in proteomics studies is an analysis of post-translational modifications, which play a crucial role in enzyme regulation, protein integration, and molecular mechanism evaluation, and which will be discussed in the next section.

### 3.1. Post-Translational Modifications (PTMs) of Proteins

Post-translational modifications (PTMs) change the structure, stability, and protein–protein interactions by the covalent adding of a functional group to the protein substrates. Psoriasis is an inflammatory skin disease characterized by keratinocyte hyperproliferation and infiltration of immune cells that are subjected to various PTMs. The most common types of PTMs in psoriasis are acetylation/deacetylation, glycosylation, citrullination, and PARylation. Most reports present the importance of post-translational modifications in single protein/enzyme studies on a genetically induced model of psoriasis in mice [102,103]. Fewer reports present PTMs in the human sample [104,105], where they explain how deacetylation/acetylation and phosphorylation processes are involved in SIRT1, STAT3 activity regulation in psoriatic keratinocytes. They underline the proinflammatory cytokine IFN-γ responsibility for SIRT1 reduction and STAT3 acetylation in the pathogenesis of psoriasis. PARylation is a PTM that relies on ADP-ribose moiety in addition to the amino acid in protein catalyzed by PARP, which plays a crucial role in inflammation. It was found that PARP1 regulates the expression of NF-κB in inflammation [106]. In imiquimod-induced psoriasis, PARP1 showed an anti-inflammatory action, and its PARP1^−/−^ mice model in genetic deletion exacerbated symptoms by increasing the secretion of cytokines IL-6, IL-17, and IL-23 [103]. The next common and abundant PTMs is protein glycosylation, proceed by glycosyltransferases. FUT8, one of the major glycosyltransferases generating a core fucose structure on N-glycans. In skin tissues affected by psoriasis, expression of FUT8 was upregulated and correlated with disease severity. Target identification of the FUT8 and associated proteins that regulate cell proliferation was analyzed by liquid chromatography with tandem mass spectrometry-based protein identification. FUT8 increased the EGFR fucosylation activating keratinocyte proliferation and psoriasis development [107]. Another study of serum analysis identified 12 different N-glycan, where 4 N-glycan were significantly increased in psoriasis patients compared to healthy subjects, and another 4 were decreased. The level of one N-glycan was increased gradually with the disease severity and could be a promising biomarker in early diagnosis of psoriasis and be exerted in progression monitoring [108]. The above findings suggest that glycosylation can be involved in the pathogenesis of psoriasis, but enlarged patient groups are needed for clinical validation. Several reports try to identify new types of PTM and explain their role in psoriasis. As an example, the Lysine 2-hydroxyisobutyrylation (Khib) modification was compared between lesional and nonlesional psoriasis patients. Proteomics was used to verify 72 proteins with upregulated Khib and 44 proteins with downregulated Khib [109]. Another newly reported reversible PTM is palmitoylation, in which a 16-carbon palmitoyl group is transferred onto the protein target by palmitoyl transferases and depalmitoylating enzymes. Recently, Zhou et al. investigated the role of a palmitoyl transferase ZDHHC2 in psoriasis development in IMQ-induced psoriasis mouse models [110].

Yang and Yan demonstrated the involvement of post-translational modifications, such as acetylation, phosphorylation, and glycosylation, in the pathogenesis of psoriasis. However, the interdependence between these differential PTMs has not yet been fully explained. They also underly the importance of SIRTs and PARPs regarding their therapeutic application in skin diseases. SIRTs and PARP inhibitors or activators could be a new biological drug in psoriasis development [111]. However, some effort must be taken to design inhibitors and activators specific to one target. Most of the presented reports were performed on the psoriasis mouse model using flow cytometry, western blot, or immunofluorescence to confirm the evidence of PTMs. Within recent years, mass spectrometry has proven to be extremely useful in PTMs discovery. High-resolution mass spectrometry provides a series of analytical functions that are useful for the characterization of modified proteins. Mass spectrometry-based protein identification was used in PTMs in psoriasis, as presented above [107,108,109].

### 3.2. Biological Material Selection for Proteomics/Metabolomics Studies

The selection of biological material for proteomic and metabolomic analysis in skin diseases depends on the research purpose, such as biomarker discovery, disease progression, metabolite turnover, or drug response therapies. The most recent biological material types and their complexity in proteomic/metabolomic study are presented in Table 2. Each type has different sample complexity, collection, and preparation procedure to overcome to obtain appropriate results. Plasma/serum is the most popular biological sample for proteomic/metabolomic studies in psoriasis (Ps) and psoriasis arthritis (PsA), especially when investigating disease progression and conversion of Ps to PsA [78]. Blood samples collected using the standard clinical procedure are the most useful biological material because of their complexity and because they provide data about organism’s whole metabolism. In some cases, such as psoriasis/psoriasis arthritis, analysis of blood components could be useful, as changes in white blood cells give information about inflammatory reactions and increased keratinocyte proliferation. This was found to be crucial in lipidomic studies of samples from psoriatic patients, where proteome and lipidome, skin, and its individual cell types, as well as blood and blood cells, were analyzed [85]. The advantage of a blood sample is reproducible and easy to compare between the samples from different stages of the disease progression. The disadvantage that could have an impact on the results is a hemolyzed sample from problematic blood collection or a lipemic sample from patients with hypercholesterolemia. Another important biological material in skin disease is skin analysis, which provides information on local changes caused by psoriasis [29,77,112]. Skin sample collection exists as a standard clinical procedure, such as a biopsy, blister fluid, or scraping. Depending on the method used, the tissue size, dip, and skin layers will be different, which has a significant impact on the results. Identical patient skin samples are difficult to collect, causing a problem with gathering a representative group of patients. Several proteomic studies analyzed pooled psoriasis skin samples to compare with normal-looking skin [87,113,114,115], differentiating protein expression. Some studies present proteomic profiling on the epidermis skin layer [75] or stratum corneum of diseased skin [116,117]. Comparison between the studies is more complex. Another inconvenience is the possibility of sample contamination by hair or cosmetics/drugs residue, which, when analyzed by highly sensitive mass spectrometry methods LC/MS/MS, could affect results. The biological materials mentioned above, namely skin samples and blood samples, are commonly used in the standard analysis of skin diseases. In contrast, urine sample analysis is given relatively less attention and is mainly used for drug treatment and toxicity monitoring. However, urine analysis can provide valuable insights into cell growth and division, metabolite turnover, and immunomodulatory effects. [118,119,120]. Simultaneous analysis of blood and skin samples on different disease stages will play a crucial role in understanding the functional changes in psoriasis development [29,112,121]. Sample collection, processing procedure, and analytical method validation must be followed by clinical validation on large groups of control and psoriasis patients on different disease progression stages and subtypes.

After the discovery stage, the preclinical verification and validation of putative biomarkers are needed. Identifying robust biomarkers as representative of various clinical psoriasis phenotypes would allow patients to be stratified into subgroups and tailor treatments according to personalized medicine. Over the past 20 years, several in vivo mouse psoriasis models, as well as proteomic and metabolomic studies, have contributed to the understanding of psoriasis development and the underlying pathogenesis of the disease. This has an impact on the increase in treatments for inflammatory and autoimmune diseases as it targets biological drugs in clinical care. FDA and EMA approved multiple targeted therapies, but because of heterogeneity in efficacy and tolerability, not all patients improve. To improve treatment response, a fast and precise diagnostic is needed, which is why many clinical trials still evaluate the utility of the biomarker in diagnosing psoriasis, its severity, and drug response.

## 4. Clinical Trials in Proteomic and Metabolomic Biomarkers Discovery

This review presents selected, completed, and currently recruiting clinical trials from the last decade (Table 3). The first five clinical trials completed from 2013 to 2019 were conducted to search for potential biomarkers to predict the effectiveness of treatment, development, and severity of psoriasis and psoriatic arthritis. The study group characteristics were similar in all clinical trials, including patients over 18 years old and of both genders. Of the 5 studies, 2 were conducted in Europe and enrolled nearly 100 participants each; the other three trials were conducted in the United States and had around 30 participants each.

Many of the selected clinical trials focused on biomarkers that could have the ability to predict responses to treatment with biologic disease modifying antirheumatic drugs (DMARDs), such as secukinumab, apremilast, and adalimumab. The primary purpose of the trials was to investigate inflammatory pathways and report changes in the levels of inflammatory markers at baseline and after treatment. To achieve the assumed goal, various analytical methods were used, such as RT-PCR and qPCR to assess the gene inflammatory biomarkers, flow cytometry to reveal the aberrant inflammatory profiles of cells, and histological examination to evaluate the reversing of lesion skin inflammation. The association between putative biomarkers and the progression and severity of psoriasis and psoriatic arthritis was investigated in several trials. Three independent clinical trials have identified matrix metalloproteinase (MMP-3) as a potential biomarker. This finding is consistent with the results presented by Ramessur et al., who identified MMP-3 as a proteomic candidate for predicting the development of psoriatic arthritis (PsA) from psoriasis lesions based on an analysis of 181 scientific articles [100]. Serum level of cytokines implicated in the Th17 pathway was measured in three clinical trials, indicating that IL 17A was a candidate for predicting psoriasis severity. In the study ‘Identification of New Prognostic Markers in Psoriatic Arthritis’, the concentration of IL 17A and other cytokines was correlated with markers of bone remodeling, identifying the molecular pathways involved in psoriatic arthropathy. The progression of psoriasis leads to the development of many comorbidities other than psoriatic arthritis, such as metabolic syndrome and cardiovascular diseases [122]. The study ‘Psoriasis Inflammation and Systemic Co Morbidities’ was intended to explore the pathophysiology of psoriasis and its comorbidities, but it also provided guidance on how long-term treatment of inflammation can reduce or prevent cardiovascular events.

The next five selected clinical trials presented in Table 3 are currently recruiting patients for candidate biomarkers investigation to predict severity and responsiveness to treatment in psoriasis and psoriatic arthritis. These open-label trials are multicentered, and a large number of patients are planned to be enrolled. The smallest estimated number of participants is 50 patients in the ‘An Explorative Psoriasis Biomarker Study’, conducted in the Netherlands. The clinical trial requires fewer participants than others currently recruiting. However, it assumes a very wide breadth of its outcome. The study involves the analysis of various chemokines, cytokines, and immune cells in the blood and lesion and non-lesion skin, determination of skin and fecal microbiome, analysis of epidermal homeostasis and immune cell infiltration, measurement of skin surface biomarkers concentration and stratum lipidomic analysis of corneum and biopsy transcriptome.

‘Identification of New Biomarkers to Promote Personalized Treatment of Patients With Inflammatory Rheumatic Diseases’ is a multicentered study in Denmark with the largest number of participants. A total of 11 centers across Denmark are recruiting, with 20,000 patients with psoriasis, psoriatic arthritis, and other rheumatic diseases required. The incidence of psoriasis in Denmark is 2.26%, which is higher than in other western European countries. Therefore, implementing this clinical trial is highly important and will help increase the effectiveness of personalized treatment [123]. Presented clinical trials are not limited to patients with psoriasis and psoriatic arthritis; they are open to all patients with inflammatory rheumatic skin diseases. The study ‘Metabolic Profiling of Immune Responses in Immune-mediated disease’ is conducted in the United States and is recruiting all patients with primary immunodeficiency, atopic dermatitis, and psoriasis.

Psoriasis is strongly associated with Hashimoto’s disease and shares common metabolic pathways in its pathogenesis [124]. This issue became the focus of a clinical study, ‘Metabolic Biomarkers in Hashimoto’s Thyroiditis and Psoriasis’, that was conducted in Greece. The main aim of these clinical trials is to identify metabolic biomarkers and investigate the role of epigenetic factors involved in metabolic pathways. For this purpose, the concentration of organic and fatty acids of patients with Hashimoto’s disease and psoriasis and healthy participants is planned to quantify by gas chromatography with mass spectrometry.

It should be highlighted that the completed clinical trials were mostly characterized by a limited number of participants, and the largest of them had only 100 subjects, while the ongoing recruitment for the trials estimates the number of participants to be 200–300 and even 20,000 subjects. The presented studies show that attempts to research candidate biomarkers take into account a variety of biological materials, not only serum or plasma, but also urine, synovial fluid, or skin biopsies. The broad extent of measurements, methods and multifarious characteristics of study groups provide the increased clinical value and usefulness in clinical practice to the biomarkers.

## 5. Conclusions

In vivo psoriasis mouse models do not fully reflect the human pathophysiology of psoriasis. However, they can be a powerful tool for preclinical application and pick out biomarkers for future research in human psoriasis. Non-targeted mass spectrometry analysis is mainly used by scientists to discover and explain some molecular mechanisms of skin disease, thus proposing the possible therapeutic target and monitoring of treatment. However, this is usually performed on small patient groups and needs to be validated during longitudinal clinical trials. The availability of multiple biological therapies did not give the expected results in all patients because of heterogeneity in efficacy and tolerability. That is why untargeted proteomics should be available for patients that did not respond to treatments. Identifying robust biomarkers as representative of various clinical psoriasis phenotypes would allow the stratification of patients into subgroups and treatments to be tailored to them as personalized medicine. The most significant conclusion is that the majority of presented psoriasis-like dermatitis in vivo models, academic research, and clinical trials focus on intensive research on proteomic and metabolic markers in psoriasis diagnosis, their severity, and in drug response treatment. The use of multiomic technologies is currently essential and is one of the most promising directions in the identification of biomarkers associated with psoriasis and psoriatic arthritis.

## Figures and Tables

**Table 1 ijms-24-09507-t001:** Rodents as a model for psoriasis.

Model Type	Animal	Most Common	Mechanism of Psoriasis Development	Advantages	Disadvantages	Ref.
Induced modulation of skin environment	Mice Rat	Imiqumod induced	Activation of: Toll-like receptor 7 ligand, Macrophages monocytes	Inexpensive Easy-to-use Commercial available drug for induction	Acute skin inflammation model Limited to skin only (without arthritis and cardiovascular disease) Different respond in different mice strain	[24,26,70,71]
		Dermal injection of IL23	Stimulation of production IL-19 and IL-24	Easy-to-use Inexpensive	Non-selective disease induction Acute skin inflammation model Convenient for drug screening	[23,32]
Xenograft model	Immunodeficient mice	Human skin transplant to AGR129 mice, C.B-17 SCID, nOG and hIL-2 NOG scid/scid mice	Xenograft possesses human immune cells which affect recipient and can induce skin lesions.	Reflects psoriasis in humans Devices for drug development T-cell activity	Limited availability of human material Decrees T-cell activity during timeline Limited use in exploration of pathophysiological aspects of systemic inflammation, immune cells response, and their correlation Expensive model Technically challenging Genetic properties of the mouse affect model	[36,38,40,44,72]
T-cell transfer	Immunodeficient mice Immunocompromised non-transgenic rat	CD4^+^/CD45RB^hi^ T-cell transfer T-cell transfer from HLA-B27 rats	Shifting the balance of immune cells	Tool to explore inflammation in psoriasis Potent instrument to identify pathways involved in immune cells response	Difficult to use Affected by genetic properties of the recipient	[56,59,60,61]
Spontaneous mutation	Mice	Asebia (AB) mice flaky tail mice flaky skin (Fsn) mice	mechanism dependent on the specific gene(s)	Histopathological correspondence to the human psoriasis (acanthosis, infiltration of mast cells and macrophages, keratocyte hyperproliferation)	Lack of T-cells in infiltrates Weak response to antipsoriatic treatment	[13,14,15,16,19]
Transgenic models	Mice Rats	Gene overexpression of, e.g., TGFα, IL-6, INFγ, VEGF, IL-23, MEK1; KLK6^+^ Gene inactivation: CD18^−/−^, IL-1RA^−/−^, integrin αE^−/−^, and many others	mechanism dependent on the down/upregulation of the specific gene(s)	Chronic skin inflammation model which has similarities with psoriasis Development of arthritic inflammation Powerful model to study pathogenesis Can be controlled via tamoxifen Can be tissue-specific	Changes in single gene as standard Lack of full histological and clinical symptoms Expensive	[54]
Genome editing	Mice	CRISPR/Cas9 technology	mechanism dependent on the down/upregulation of the specific gene(s)	Precision of gene editing Changes in multiple genes at the same time Less time needed to generate transgenic mouse Powerful model to study pathogenesis	Expressive Off-target biological phenotype	[62,64,67]

TGF-α transforming growth factor α; IL-6—interleukin 6; IL-19—interleukin 19; IL-23—interleukin 23; IL-24—interleukin 24; INF-γ -interferon gamma; VEGF—vascular endothelial growth factor; KLK6 -, kallikrein-related peptidase 6; CD18—integrin beta chain-2; IL-1RA—interleukin-1 receptor antagonist; CRISPR/Cas9—clustered regularly interspaced short palindromic repeats associated protein 9; CD+/CD45RBhi T-cells—naive T cells from B10.D2 mice.

**Table 2 ijms-24-09507-t002:** Biological material types and their complexity in proteomic/metabolomic analysis in skin disease.

Sample	Skin	Plasma/Serum	Urine
Easy to: ○ collect ○ transport ○ store	+ +++ +++	++ +++ +++	+++ +++ +++
Non-invasive	+	++	+++
Reproducible	+	+++	++
Complexity	++	+++	+
Preparation	++	+++	++
Metabolomics	++	+++	+
Proteomics	++	+++	+
Contamination	hair cosmetic/drugs residues	hemolyzed lipemic	proteins particles
Other	Different method for sample collection: biopsy, blister fluid, scraping, epidermis, stratum corneum	-	-

**Table 3 ijms-24-09507-t003:** Selected, completed, and currently recruiting clinical trials, based on https://clinicaltrials.gov/ (accessed on 6 February 2023).

Clinical Trial/ NCT Number	Localization	Number of Participants	End-of-Clinical Trial	Biomarkers Examined	**Method**
**Psoriatic Inflammation Markers Predictive of Response to Adalimumab (IMPRA)/** **NCT03389984**	France	85	2019	mRNA expression in skin; Drug response: Adalimumab	RT-PCR
**Multiple-dose Regimen Study to Assess Effect of 12 Months of Secukinumab Treatment on Skin Response and Biomarkers in Psoriasis Patients/** **NCT01537432**	United States	36	2014	histological disease reversal score in skin biopsies; Drug response: Secukinumab	histological examination
**Monocyte Biomarkers in Moderate to Severe Plaque Psoriasis Subjects Treated With Apremilast/** **NCT03442088**	United States	28	2018	Serum markers of activated monocytes, monocyte transcriptome biomarkers, serum MPO, resistin, IL-17, and TF; Drug response: Apremilast	flow cytometry, qPCR
**Psoriasis Inflammation and Systemic Co Morbidities/** **NCT01170715**	United States	29	2013	IL17A gene expression, inflammatory cytokines, metabolic and vascular markers; Drug response: Etanercept	ECL, RT-PCR, histological examination
**Identification of New Prognostic Markers in Psoriatic Arthritis/** **NCT03455166**	Italy	100	2016	Serum levels of cytokine referable to Th17 pathway, MMPs, TIMPs, and markers of bone remodeling.	Data not provided
**An Explorative Psoriasis Biomarker Study/** **NCT04394936**	Netherlands	50	2022	Lipidomic of the stratum corneum, cutaneous and fecal microbiome, immune cell subsets and protein biomarkers, and genotyping Drug response: Guselkumab	LC-MS, ELISA, NGS, flow cytometry
**Metabolic Biomarkers in Hashimoto’s Thyroiditis and Psoriasis/** **NCT04693936**	Greece	200	2024	Urinary organic acids levels, blood fatty acids	GC-MS
**Identification of New Biomarkers to Promote Personalized Treatment of Patients With Inflammatory Rheumatic Diseases/** **NCT03214263**	Denmark	20,000	2024	Data not provided	Data not provided
**Molecular Signatures in Inflammatory Skin Disease (MSID)/** **NCT03358693**	Germany	300	2028	Immune cell composition, transcriptome, proteome, and microbiome signatures	DNA/RNA sequencing, ELISA, MS, flow cytometry
**Metabolic Profiling of Immune Responses in Immune-mediated diseases/** **NCT04864886**	United States	150	2027	Metabolomics of immune-mediated diseases in peripheral blood cells, serum, skin biopsies, skin tape strips, and skin swabs, metabolomics of the microbiome	RNA sequencing, MS

RT-PCR—reverse transcription polymerase chain reaction, qPCR—quantitative polymerase chain reaction, ECL—electrochemiluminescence, MPO—myeloperoxidase, TF—tissue factor, IL-17—interleukin 17, MMPs—matrix metallopeptidase, TIMPs—tissue inhibitor of metalloproteinase, LC-MS—liquid chromatography-mass spectrometry, ELISA—enzyme linked immunosorbent assays, NGS—next-generation sequencing, GC-MS—gas chromatography-mass spectrometry, MS—mass spectrometry.

## Data Availability

Data sharing not applicable.

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
