# Peer review of "Proteomic and Metabolomic Changes in Psoriasis Preclinical and Clinical Aspects"

_ijms, 2023, doi:10.3390/ijms24119507_

Round 1

Reviewer 1 Report

None

Author Response

Thank you for your kind assessment.

Reviewer 2 Report

Radulska et al presented a manuscript on proteomic and metabolomic changes in skin diseases, in particular psoriasis. There are a few suggestions which might help improve the manuscript. 

1. The focus of the manuscript should be proteomics and metabolomics studies. Large part of the manuscript on psoriasis mouse models seem less relevant to the main theme. 

2. Intensive proofreading should be performed as it is quite difficult to read through the manuscript.

3. Since the entire manuscript is talking about psoriasis, it might make more sense if "selected skin diseases" in the title can be replaced with just "psoriasis".

Author Response

1. The focus of the manuscript should be proteomics and metabolomics studies. Large part of the manuscript on psoriasis mouse models seem less relevant to the main theme. 

Thank you for your kind suggestion. We propose corrected Title:

 “Proteomic and metabolomic changes in psoriasis preclinical and clinical aspects.”

2. Intensive proofreading should be performed as it is quite difficult to read through the manuscript

We have thoroughly reviewed the revised text of manuscript.

3. Since the entire manuscript is talking about psoriasis, it might make more sense if "selected skin diseases" in the title can be replaced with just "psoriasis".

Thank you for your kind suggestion. We propose corrected Title:

 “Proteomic and metabolomic changes in psoriasis preclinical and clinical aspects.”

Reviewer 3 Report

Very interesting and topical work.

In this article, the authors outline the animal models, biological samples and proteomics and metabolomics results for the study of psoriasis currently presented in the literature.

I suggest some corrections:

1) The title should be revised: "selected skin disease" lets think of a group of diseases, whereas in the text only psoriasis and psoriatic arthritis are treated, and frequently not even that well differentiated. Therefore, in fact, this boils down to the treatment of only one skin disease: psoriasis. Perhaps the title should be replaced with 'preoteomic and metabolomic changes in psoriasis'.

2) In the introduction it would be useful to mention, among others, the 2021 paper published in the Lancet: Griffiths CEM, Armstrong AW, Gudjonsson JE, Barker JNWN. Psoriasis. Lancet. 2021 Apr 3;397(10281):1301-1315. doi: 10.1016/S0140-6736(20)32549-6. PMID: 33812489. Please, consider it especially for the clinical manifestations part.

3) In lines 62-64, the clinical description needs more detail. It is important to describe psoriasis plaque (with erythema, infiltration, desquamation). The sites of psoriasis should be reviewed, e.g. include elbows and knees, which are very characteristic. Interesting to mention, even quickly, the Auspitz's sign and Koebner's phenomenon.

4) In Chapter 2 it is useful to mention the work of Gangwar RS, Gudjonsson JE, Ward NL. Mouse Models of Psoriasis: A Comprehensive Review. J Invest Dermatol. 2022 Mar;142(3 Pt B):884-897. doi: 10.1016/j.jid.2021.06.019. epub 2021 Dec 23. PMID: 34953514.

5) It is necessary to explain to readers how the articles were searched and by which methods the literature was reviewed and selected: search engines, keywords, search strings, etc.

6) The conclusions can be expanded. You can explain what the usefulness of proteomics and metabolomics studies is, detail the diagnostic and therapeutic perspectives. You can explain the usefulness of these studies in the evaluation of differences between patients, in order to adopt more effective and targeted therapies and with a view to increasingly personalised medicine, for example. You should spend a few words on the future perspectives of proteomics and metabolomics.

The tables and pictures are clear. English is fluent.

Overall the work is good and summarises elements of great interest.

Author Response

1) The title should be revised: "selected skin disease" lets think of a group of diseases, whereas in the text only psoriasis and psoriatic arthritis are treated, and frequently not even that well differentiated. Therefore, in fact, this boils down to the treatment of only one skin disease: psoriasis. Perhaps the title should be replaced with 'preoteomic and metabolomic changes in psoriasis'

Thank you for your kind suggestion. We propose corrected Title:

 “Proteomic and metabolomic changes in psoriasis preclinical and clinical aspects.”

2) In the introduction it would be useful to mention, among others, the 2021 paper published in the Lancet: Griffiths CEM, Armstrong AW, Gudjonsson JE, Barker JNWN. Psoriasis. Lancet. 2021 Apr 3;397(10281):1301-1315. doi: 10.1016/S0140-6736(20)32549-6. PMID: 33812489. Please, consider it especially for the clinical manifestations part.

We will add the reference.

3) In lines 62-64, the clinical description needs more detail. It is important to describe psoriasis plaque (with erythema, infiltration, desquamation). The sites of psoriasis should be reviewed, e.g. include elbows and knees, which are very characteristic. Interesting to mention, even quickly, the Auspitz's sign and Koebner's phenomenon.

We will add more clinical details.

4) In Chapter 2 it is useful to mention the work of Gangwar RS, Gudjonsson JE, Ward NL. Mouse Models of Psoriasis: A Comprehensive Review. J Invest Dermatol. 2022 Mar;142(3 Pt B):884-897. doi: 10.1016/j.jid.2021.06.019. epub 2021 Dec 23. PMID: 34953514.

Thank you for your kind suggestion. As a rule, we tried to refer to original works. We added the above reference, at your request.

5) It is necessary to explain to readers how the articles were searched and by which methods the literature was reviewed and selected: search engines, keywords, search strings, etc.

Thank you for your kind suggestion, we followed the editor suggestion for review papers recommendation. We added more detail below.

Electronic literature searches were performed in PubMed and NIH U.S National Library of Medicine ClinicalTrials.gov on January 2023 for studies in the English language, published within recent 5 years and older (for original papers of mouse models). Main used keywords: proteomic/metabolomic in psoriasis, psoriasis development, prosiasis diferentation, mouse models in psoriasis, psoriasis types, epidemiology of psoriasis, psoriasis biomarkers, proteomic/metabolomic in skin disease. We considered mainly original papers with few summarized review.

6) The conclusions can be expanded. You can explain what the usefulness of proteomics and metabolomics studies is, detail the diagnostic and therapeutic perspectives. You can explain the usefulness of these studies in the evaluation of differences between patients, in order to adopt more effective and targeted therapies and with a view to increasingly personalised medicine, for example. You should spend a few words on the future perspectives of proteomics and metabolomics.

We have thoroughly reviewed the revised text.

Round 2

Reviewer 2 Report

The authors have been in a good shape for publication.